# Emergency Department Syndromic Surveillance to Monitor Tick-Borne Diseases: A 6-Year Small-Area Analysis in Northeastern Italy

**DOI:** 10.3390/ijerph20196822

**Published:** 2023-09-25

**Authors:** Massimiliano Colucci, Marco Fonzo, Liana Miccolis, Irene Amoruso, Sara Mondino, Andrea Trevisan, Romina Cazzaro, Tatjana Baldovin, Chiara Bertoncello

**Affiliations:** 1Hospital Direction, Local Health Authority 8 (Azienda ULSS Berica), Veneto Region, 36100 Vicenza, Italy; 2Hygiene and Public Health Unit, Department of Cardiac Thoracic Vascular Sciences and Public Health, University of Padova, 35131 Padova, Italy

**Keywords:** syndromic surveillance, tick-borne diseases, Lyme disease, tick-borne encephalitis, incidence, epidemic intelligence, small-area analysis, emergency department, Italy, tick-bites

## Abstract

Tick-borne diseases (TBD) are endemic in Europe. However, surveillance is currently incomplete. Alternative strategies need to be considered. The aim of this study was to test an Emergency Department Syndromic Surveillance (EDSyS) system as a complementary data source to describe the impact of tick bites and TBD using a small-area analysis approach and to monitor the risk of TBD to target prevention. ED databases in the Local Health Authority 8 District (Veneto, Italy) were queried for tick-bite and TBD-related visits between January 2017 and December 2022. Hospitalisations were also collected. Events involving the resident population were used to calculate incidence rates. A total of 4187 ED visits for tick-bite and 143 for TBD were recorded; in addition, 62 TBD-related hospitalisations (of which 72.6% in over 50 s and 22.6% in over 65 s). ED visits peaked in spring and in autumn, followed by a 4-week lag in the increase in hospital admissions. The small-area analysis identified two areas at higher risk of bites and TBD. The use of a EDSyS system allowed two natural foci to be identified. This approach proved useful in predicting temporal and geographic risk of TBD and in identifying local endemic areas, thus enabling an effective multidisciplinary prevention strategy.

## 1. Introduction

Tick-borne diseases (TBD) are endemic in Europe. Among them, Lyme borreliosis (LB) and tick-borne encephalitis (TBE) have the highest incidence and impact on the population. Although LB is the most common TBD in Europe, the incidence of TBE has increased in recent years [1,2,3,4,5,6]. LB is caused by spirochetes of the Borrelia burgdorferi sensu lato complex (*Spirocheatales*, *Spireochaetaceae*), whereas the causative agent of TBE is the tick-borne encephalitis virus (*Flavivirus*, *Flaviviridae*). Both diseases are zoonoses, maintained in more or less geographically delimited areas in nature, called “natural foci”. Humans are accidental, mostly dead-end hosts, who become infected when they enter the natural foci [7,8]. Detailed knowledge of the spatial and temporal distribution of infection risk is necessary to take effective measures to limit the exposure of human populations to vector-borne pathogens. However, surveillance of LB and TBE in Europe is currently incomplete, which means that the available incidence data most likely only partially reflect the true risk and thus underestimate the phenomenon [1,3,9,10,11].

The incidence of LB in Europe is uncertain [1]. According to Burn and colleagues, four countries have an incidence of LB higher than 100 per 100,000 person-years, while most of countries report less than 20 cases per 100,000 person-years; however, as the authors themselves note, comparisons of incidence between countries must be made with caution because of heterogeneity in study design and methods, including non-negligible variability in the definition of LB cases [4]. In addition, several countries reported high incidence of LB at the subnational level, suggesting that national estimates may not be an effective tool for monitoring the true impact of LB. As reported in a recent review, the seroprevalence of LB in Europe varies between 0–70% [5]; this wide gap may reflect a real difference in distribution, but it should be kept in mind that differences in the epidemiologic methods used may significantly affect the possibility of comparison [4,5,12]. For instance, some surveillance systems include only laboratory-confirmed cases, while others consider also probable or suspected LB cases. The accuracy of epidemiologic data is affected by inconsistent case definitions, inconsistent diagnostic procedures, and the lack of a shared quality management system [1]. The limitations of both surveillance systems and epidemiologic studies are likely to contribute to an underestimation of the incidence of LB in Europe [4]. Furthermore, as highlighted in a recent review, there is still great heterogeneity in national surveillance systems and LB control policies across European countries [13]. Without a common indicator, it is difficult to obtain a clear epidemiologic picture.

Alongside these issues, it must be considered that the burden of LB in Europe is expected to increase as the tick population is expanding, possibly as a results of the impact of climate change on tick life cycles, migratory animals, and human activities [14,15,16]. This expansion is evident for TBE [17]. The cumulative number of reported TBE cases in 2020 was double that in 2015, with an increasing trend also in regions previously thought to be free of the responsible virus, following a north-westward spread in continental Europe [2]. Indeed, an increase in reported TBE cases was observed in almost all European countries, with Germany, Italy, Norway, Slovenia and Switzerland reporting a more than threefold increase between 2015 and 2020 [2,3,9]. It is almost certain that documented cases of TBE in Europe represent a relatively small proportion of the true burden of infection: in fact, (i) the EU case definition includes only clinically relevant cases with a laboratory-confirmed diagnosis—on the contrary, most TBE infections are asymptomatic, and mild forms of the disease may go undiagnosed; (ii) some countries have incomplete epidemiologic information; (iii) some countries use a slightly different case definition; (iv) countries have implemented different surveillance systems and testing practices [2]. In Italy, the notification system is based only on cases with clinical signs of central nervous system inflammation (e.g., meningitis, meningoencephalitis, encephalomyelitis, encephaloradiculitis) [18]. Surveillance for TBD usually focuses on well-defined areas considered to be at risk, with the aim of implementing effective risk management plans, as TBDs tend to be geographically restricted. However, monitoring activities should not be restricted to endemic areas, as limited surveillance does not allow early identification of new endemic areas, thus increasing the risk of infection in the population. In addition, restricting attention to high-risk areas may limit awareness and reduce sensitivity in the identification and diagnosis of TBD in areas considered non-endemic [1,3]. Underreporting and resource limitations have had a significant and negative impact on the accuracy of TBE incidence surveillance. In view of this, alternative surveillance strategies for TBE need to be considered [19,20,21,22,23,24].

Because many people may seek and receive care for tick bites that expose them to the risk of TBD, some countries have begun to consider data on tick bites as well as disease [13]. The information on tick bites can be useful both to highlight areas where the phenomenon is more common and to monitor over time the ability of health systems to reduce the incidence of bites and thus the risk of contracting TBD.

Public health agencies in the United States have implemented emergency department syndromic surveillance (EDSyS) systems to collect data for early detection of LB. The system monitors tick-bite-related consultations and not just those associated with LB, and serves as a source of tick-bite risk information to inform prevention messaging and health worker education. In Europe, the Netherlands and Switzerland monitor the number of medical consultations for tick bites and erythema migrans; in Belgium, France, Liechtenstein, the Netherlands, and Switzerland, a national government agency is involved in a public participation website or app [13].

Italy is considered a low-incidence country for both LB and TBE, with an endemic circulation recorded only in some areas in the North-East. As in other European countries, Italian data largely underestimate the true burden of LB and TBE, especially in endemic areas [10,11,25,26]. This limits the ability to monitor the extent of the problem, both in terms of the number of people affected and the geographical extent. In Italy, however, administrative health data are usually available and can be a rich source of information, also from a public health point of view. Hospital discharge records have been the most commonly used data source for TBD identification. However, data on emergency department (ED) consultations can be an additional source of information on tick bites and TBD cases. This make it possible to align TBE prevention strategies with those for LB, since most of the actions are identical, especially those aimed at preventing bites or removing ticks, e.g., At present, a joint preventive approach to the two diseases is rare, but where it has been used, encouraging results have been achieved in controlling both diseases [27].

The aim of this study was to test an EDSyS system as a complementary data source (1) to describe the impact of tick bites and tick-borne diseases (TBE and LB) at a high level of geographical resolution using a small-area analysis approach and (2) to monitor the risk of tick-borne diseases, in order to target educational messages for their prevention.

## 2. Materials and Methods

The study was conducted in the district of the Local Health Authority (LHA) 8 (Azienda ULSS Berica) Veneto region, Northeastern Italy, which provides health care for approximately 500,000 people and covers 59 municipalities. All four acute hospitals were included and data on emergency department visits and hospital admissions were collected.

A syndromic surveillance approach was adopted. Syndromic surveillance is a form of surveillance that generates information to drive public health action by collecting, analysing and interpreting routine health-related data on symptoms and clinical signs reported by patients and clinicians, rather than relying on microbiologically or clinically confirmed cases [28]. The intent of syndromic surveillance as an early warning system for bioterrorism has expanded to an all-hazards surveillance approach that provides real-time information for situational awareness, characterisation of health events and response efforts [29]. The method used in the current study was inspired by the work of Daly and colleagues [19], with some adaptations for the Italian context and the type of data available.

ED databases were queried for tick-bite and TBD-related visits between January 2017 and December 2022. Data at the ED record level included the following key variables: age, sex, residence, date of ED visit, International Classification of Diseases 9th Revision (ICD-9-CM) diagnosis code, and free text annotations by the attending physician. The free text was also searched for words related to tick bites, erythema migrans, LD and TBE, including their common misspellings. The ICD-9 codes used were: “063”, “088.81”. ED consultations were categorised as tick-bite-related visits *(ED-bite)* or as TBD-related visits *(ED-TBD)*. *ED-bite* were those without a TBD-related ICD-9 diagnosis code or a TBD suspicion in the medical free text annotations. *ED-TBD* were those with a confirmed or suspected TBD either as an ICD-9 code or in the free text. After a computerised extraction, two researchers independently reviewed the ED records to assess whether the inclusion criteria were met.

In addition, we considered hospital discharge records from the same hospitals during the same time period (January 2017–December 2022). Inpatient-level data included the following key variables: medical record number, age, sex, city and state of residence, date of admission and discharge, ICD-9 diagnosis code. TBD-related hospitalisations *(hos-TBD)* were those with the following ICD-9 codes: “063”, “088.81”. Data were analysed to calculate frequencies for key demographic and reporting characteristics using Microsoft Excel (Redmond, WA, USA). Overall incidence rates were calculated using population data from the Italian National Institute of Statistics (ISTAT) on 1 January 2023. The smallest territorial unit considered in this study was the municipality. For each municipality, the mean elevation was obtained from the ISTAT database. The calculations are based on the spatial bases and the Digital Elevation Model (DEM) of the main spatial units of interest for official statistics. The size of the single spatial unit used to calculate the mean elevation was 20 m × 20 m [30].

## 3. Results

Between January 2017 and December 2022, EDs in the LHA 8 Berica area registered 921,282 accesses overall. Among these, a total of 4430 *ED-bite* and *ED-TBD* events were identified (Table 1).

The highest number of *ED-bite* events was recorded in 2018, although the highest number of *ED-TBD* and *hos-TBD* was in 2019. In 2020 there was a reduction in all events that started rising from 2021 on. Demographic characteristics of population involved was similar in terms of gender for all the events investigated *ED-bite*, *ED-TBD* and *hos-TBD*, with a higher proportion of males (59.5%, 62.2% and 61.3%, respectively). With regard to the age of involved population, the highest number of *ED-bite* was recorded in 0–5 and 5–10 aged, with a considerable number of *ED-TBD*, as well (Figure 1).

The risk of ED-bite is present all the year (there are no months free of *ED-bite* events). However, two peaks were registered: the highest in size was in May, followed by a second in October. The highest peak of *ED-TBD* occurred in June and the temporal trend is similar to *ED-bite* with roughly a 4-weeks delay. Speaking of hospitalisations, the peak was in July, followed by a second (and lower) one in November (Figure 2).

In order to assess the risk of tick-bite and to calculate the incidence, only ED-related events in people living in the area were considered, so the number of ED visits considered was 4055. Figure 3 shows the incidence of bites per municipality of residence (Figure 3a) and the mean elevation of the municipalities (Figure 3b). The map highlights two subareas at higher risk (municipalities with *ED-bites* incidence >7.0/1000/6-year): one in the north-west and the other in the south-west, clearly separated by municipalities where the incidence of *ED-bites* is lower than 7.0. Geographically, the two areas identified are made up of an aggregate of municipalities with the highest mean elevation among those in the LHA 8 Berica district area. In the north-western area, the mean elevation ranges from 80 to 950 m above sea level (on average 375 m), in the south-western area from 35 to 270 m above sea level (on average 100 m). The remaining municipalities have a mean elevation of less than 130 m above sea level (on average 50 m).

To assess whether the incidence of *ED-bite* events was associated with a higher risk of contracting TBD, the annual hospitalisation rate (per 100,000) for TBD (LB and TBE, separately) was determined in each of the sub-areas of interest and for the rest of the district. Of the 62 *hos-TBD* events, 52 occurred in LHA 8 Berica residents. Both sub-areas showed a higher incidence of LB than the rest of the district, although only the north-west subarea had a significantly higher incidence of TBE hospitalisations than the LHA 8 Berica district as a whole. Interestingly, there were no TBE hospitalisations in the south-west subarea (Table 2).

## 4. Discussion

The data presented in this report show that, in our context, patients frequently attend the ED for the treatment of tick bites and symptoms associated with diseases suspected to be tick-borne. To our knowledge, this is the first study conducted in Italy using EDSyS to monitor tick-borne diseases. Over the 6-year period under investigation (2017–2022), there were 4330 visits to the ED for a tick or TBD-related health concern; this volume represents 0.48% of all ED visits. Not surprisingly, more ED visits were related to tick bites than to suspected TBD, a finding observed in other studies [19,31]. Our data are consistent with data described by Daly and colleagues, who reported a number of ED visits related to a tick or LB, which represented 0.5% of all ED visits [19]. These data highlight the burden that tick-related ED visits place on our healthcare system, beyond those who develop tick-borne disease, as the majority of ED visits in this study were related to tick bites.

The gender distribution of patients seeking care at ED was similar to hospitalisations, showing a slight predominance of males, as in previous studies [2,19,32].

In terms of age, ED-visits showed higher rates in the very young (aged ≤ 10) compared to *hos-TBD* (24.0% vs. 6.4%). These differences may reflect age-related differences in ED use for health services, with young children having the highest rates of ED use [19,33].

Newitt and colleagues described the highest incidence of bite-related consultations in the 45–64 age group and in the 0–15 age group, with some differences in the use of primary care and hospital care [33]. The incidence of ED visits and calls to remote advice services was highest in children. Our results show a greater number of *hos-TBD* events in adults, 72.6% in the over 50s, including 14.5% in the over 65s. At the same time, only 39.0% of ED visits were in the over 50s, of which 18.6% were over 65, suggesting a greater perception of risk in the over 65s. Both the ED visits and the hospitalisations indicate a need for targeted TBD prevention messages to parents of young children and adults aged 50–65.

The seasonal distribution of tick-borne ED visits peaked earlier than hospitalisations in spring and autumn, consistently with previous literature [19]. The spring peak was higher than the autumn peak. Daly and colleagues reported that both LB cases and LB-related ED encounters showed a single peak in summer [19]. In contrast, our data showed a second, albeit smaller, peak in November for both LB and TBE. However, there was a difference in the seasonal distribution of hospitalisations between the two diseases. LB hospitalisations were recorded in all months of the year, whereas 30 out of 31 TBE hospitalisations (98.7%) were recorded between June and November. In addition, TBE hospitalisations, although low in number, were concentrated in spring and autumn. Periodic analyses should be planned to look more closely at seasonal patterns of ED visits to determine the optimal timing of messages on the importance of resorting to personal protective behaviours and vaccination.

The small-area analysis approach showed a clear geographical distribution of the incidence rate of *ED-bite* events per 1000 inhabitants in the municipalities (over a 6-year period), thus making possible to identify two subareas at higher risk for tick bites: a north-west subarea with an incidence rate of *ED-bite* per 1000 populations ranging between 8.3 and 46.0 and a south-west area with an incidence 7.2–34.9. The incidence rate of tick bites decreases concentrically from the municipality with the highest incidence in each subarea. In both subareas, this is the municipality with the highest mean elevation. The ED-bite 6-year incidence in the remaining municipalities is less than 7.0 per 1000. These two emerging subareas showed higher rates of *hos-TBD*. Overall, the incidence of hospitalisations is higher where the incidence of tick bites is higher. Surprisingly, both have the highest incidence of *LB-hos*, whereas *TBE-hos* were reported only the north-west subarea. There are probably barriers between the two areas that have prevented TBEV-positive ticks from moving from north to south. The border area between the two areas is a flat area crossed by major roads and railways. Further studies are needed to investigate the reasons for this observation.

Italy is considered a low risk country for both LB and TBE. According to Sykes and Makiello, the incidence of LB in Italy during 2001–2015 was 0.01 new cases per million resident population/year, but the disease is probably underdiagnosed and underreported. Currently, northern regions have a higher incidence of LB than central and southern Italy, where LB appears to be hypoendemic. However, the epidemiology and spatial distribution of LB cases are poorly understood and the regional trend of disease incidence has not been described [26]. The mean annual incidence in the Veneto region between 2015 and 2019 was 1.92/100,000 inhabitants [34]. TBE in Italy has an estimated notification rate of 0.38 cases per 100,000/year in the period 2000–2013 [10]. Endemicity for TBE is historically limited to the north-eastern regions. Veneto region in the period 2017–2020 showed a crude incidence ratio of 0.26 per 100,000 [25].

However, it is reasonable to assume that these data largely underestimate the true epidemiology, for several reasons. First, bulletins issued by the Italian National Institute of Health (ISS) only report on TBE cases characterised by meningitis and/or encephalitis, which represent only 20–30% of all TBE infections; secondly, mandatory reporting systems fail to detect a large percentage of patients (up to 45%) if hospital discharge data are not properly integrated.

In the Veneto Region, there is one local endemic area for TBE: the province of Belluno (LHA 1), which has an incidence rate of 5.89 per 100,000 (calculated for the period 2007–2017). This rate is higher than the WHO threshold of 5 cases per 100,000 for endemic areas. This is a mountainous area in the north-east, bordering Austria. An active vaccination policy for the general population is currently ongoing in this area [35].

Although our LHA district is a non-endemic territory for TBD (0.99/100,000 for LB and 0.78/100,000 for TBE) it nevertheless shows the presence of two ‘natural foci’ of disease: a first subarea in the north-west with an incidence of LB of 2.84/100,000/year (higher than the LHA district of affiliation, Veneto and Italy) and of TBE of 1.84/100,000/year (higher than the LHA district of affiliation, Veneto and Italy); and a second subarea in the south-west with an incidence of LB of 1.60/100,000/year (higher than the LHA district of affiliation and Italy) but with no admissions for TBE in the 6 years investigated. These areas have the highest numbers of patients visiting the ED for tick bites or bite-related consequences, as already demonstrated in previous studies [19,32].

The two subareas fit into a south-west trajectory with respect to the endemic area for TBE, which in Veneto is represented by the province of Belluno (LHA 1), the same direction described in continental Europe. Targeted tick control programmes could help to level out risk hotspots and raise public awareness [1,2].

A number of facts make it important to monitor these diseases in all territories, especially in the North East of Italy: the underestimation of the incidence of both diseases in Italy, the increase in the incidence involving this country, the emergence of new endemic areas, the shift in the south-west direction of traditional endemic areas. As several authors have pointed out, the current surveillance system is inadequate both in terms of its ability to identify all cases and in the rapidity of detecting new outbreaks. The adoption of a EDSyS system for the identification of bites and suspected cases is important to identify foci in non-endemic areas. This may make it possible to intervene early in order to monitor and contain exposure to risk by monitoring *ED-bite* events, prevent an increase in cases through actions targeted in time and space to the population most exposed and to doctors working in the area so that they are sensitised to the formulation of diagnostic suspicion and the notification of cases.

For the purposes of proper prevention, in fact, it is not possible to disregard a good awareness and knowledge among the general population to improve adherence to personal protection interventions, starting in March and then resuming in late summer to prevent the autumn peak; a full awareness of clinical physicians to ensure a more certain and timely diagnosis; a better knowledge of the subject on the part of public health workers in general in order to improve the timing and location of possible interventions, including vaccination.

In addition, combining hospitalisation data with bite data is useful to inform about the risk of acquiring the disease and also to highlight its severity, particularly as it has been shown that the perceived severity of a tick bite and TBD significantly predict the level of adoption protection [36].

A few strengths and limitations of our study have to be mentioned. First, there is no a shared protocol for the management of cases presenting with tick bites or for the diagnosis of LB or TBE in our EDs, as there is no a shared protocol for coding cases in the ED information system. Therefore, there may be variability in coding and suspicions. However, TBE suspects are more difficult to identify by textual search because symptoms are often non-specific. The information that proved to be most relevant in defining risk areas were *ED-bite* events, which accounted for 95 per cent of cases. In addition, in cases where TBD was diagnosed or expected to be diagnosed, the *ED-bite* event was recorded (almost all suspected cases of the disease had a previous bite reported). Second, with regard to the hospital discharge records, we did not have access to clinical records for more detailed information, then coding errors could not be excluded. Third, not all people go to the ED after a tick bite, since many remove the tick themselves or seek assistance by the general practitioner or paediatrician [19,33]. This is a limitation but also an additional reason to use this system, to educate people and clinicians about the possible risk of TBD associated with tick bites and then to take appropriate post-exposure surveillance actions. Lastly, the number of ED visits for tick bites as well as the number of hospital admissions for TBD may be affected by particular epidemic events as occurred during the COVID-19 pandemic. In Italy, indeed, during the pandemic, patients had limited accesses to ED and hospital [37]. This could explain the reduction in both ED visits and admissions in the years 2020 and 2021.

*ED-bite* events can be considered as a robust proxy for TBD [31]. The use of EDSyS may be the key to provide timely information that is not currently available through other means. The added value lies in the potential to predict TBD exposure in terms of time and place, and to guide timely and consistent public health interventions, such as timely dissemination of messages on the importance of personal protective behaviours [32]. The granularity of information that such a system can provide could help health authorities to set up specific alerts in particular geographical areas at the exact time when tick activity is increasing. Currently, preventing tick bites is the best option for risk reduction. In areas of high exposure, direct tick control may be an effective measure [38].

Our study makes it reasonable to adopt such a system also in non-endemic areas for continuous monitoring. However, diagnostic codes specific to tick bites were not available in any of the diagnostic code classification systems, including the ninth and tenth revi-sions of the International Classification of Diseases [32]. In the light of this, it may be use-ful to establish a protocol for coding cases in the ED information system by defining, for example, a unique way to record the ED-bite event.

## 5. Conclusions

In conclusion, the adoption of an EDSyS system to monitor TBDs was found to be effective, and a small-area analysis approach was essential in identifying foci at increased risk of exposure to tick bites and to compare it with the risk in the general population. This method makes it possible to monitor the risk of LB and TBE jointly, obviating the problem of under-reporting and the delay with which notification data are available. It is a method that uses existing data, which provides continuous information and which any LHA in Italy, particularly if it has hilly or mountainous terrain, could adopt to identify areas at risk for TBE. Such a method could be adopted, not only in Italy, but in all realities where ED data are accessible to public health authorities.

## Figures and Tables

**Figure 1 ijerph-20-06822-f001:**
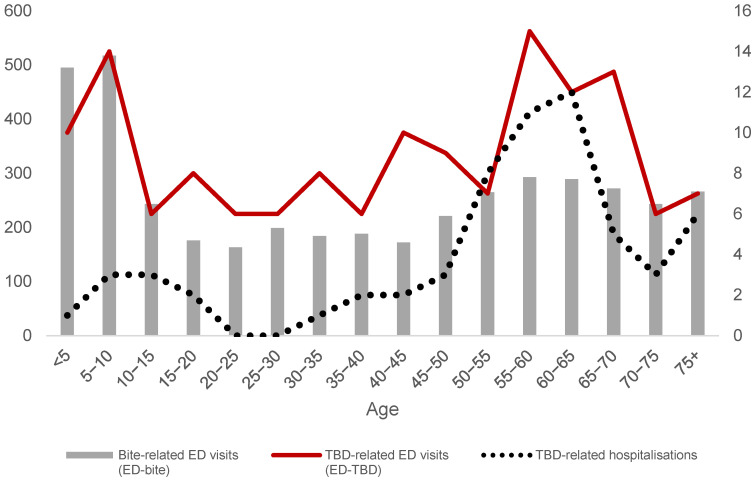
Number of bite-related ED visits (*ED-bite* events; primary *y*-axis on the left), TBD-related ED visits (*ED-TBD* events, secondary *y*-axis on the right), and overall number of TBE-related hospitalisations (*hos-TBD* events, secondary *y*-axis on the right) per age group of study population. Local Health Authority 8 (Azienda ULSS Berica), Veneto Region, Italy, 2017–2022.

**Figure 2 ijerph-20-06822-f002:**
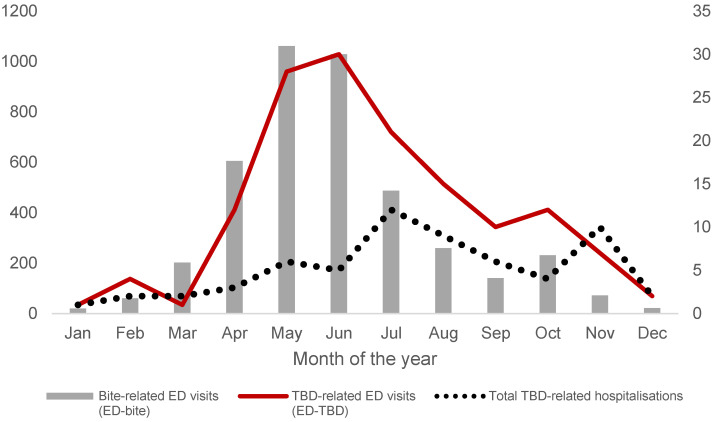
Number of bite-related ED visits (*ED-bite* events; primary *y*-axis on the left), TBD-related ED visits (*ED-TBD* events, secondary *y*-axis on the right), and overall number of TBE-related hospitalisations (*hos-TBD* events, secondary *y*-axis on the right) per month of the year. Local Health Authority 8 (Azienda ULSS Berica), Veneto Region, Italy, 2017–2022.

**Figure 3 ijerph-20-06822-f003:**
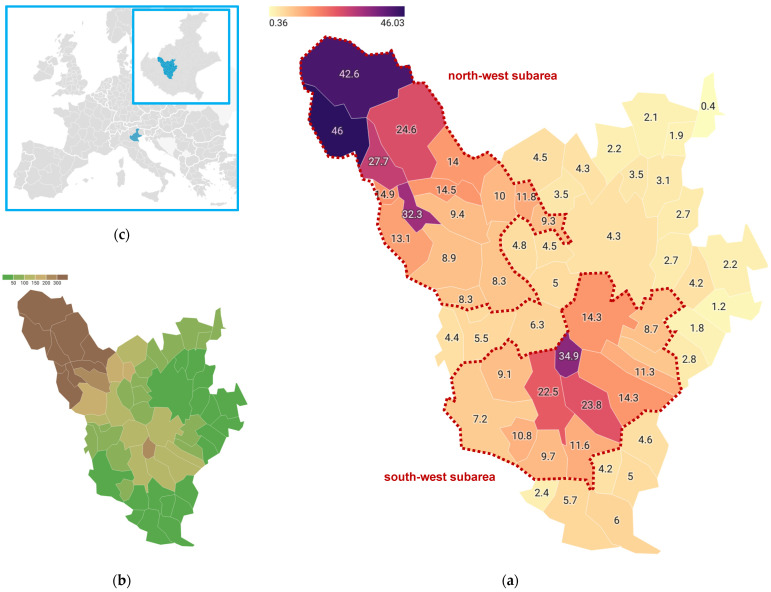
(**a**) Incidence of tick-bites (*ED-bites* events/1000 population/6-year) per municipality; municipalities with incidence >7.0 are grouped in the two subareas of interest (marked with red dotted line) (**b**) Mean elevation of municipalities in the LHA district; (**c**) Localisation of the study area. Local Health Authority 8 (Azienda ULSS Berica), Vicenza, Veneto Region, Italy, 2017–2022.

**Table 1 ijerph-20-06822-t001:** Number of bite- and TBE-related ED visits; number of LB and TBE-related hospitalisations. Local Health Authority 8 (Azienda ULSS Berica), Vicenza, Veneto Region, Italy, 2017–2022.

		Emergency Department Visits						Hospitalisations					
		Bite-Related *		TBD-Related **		Total ED Visits		LB-Related		TBE-Related		Total Hosp.	
		*n*	%	*n*	%	*n*	%	*n*	%	*n*	%	*n*	%
Total		4187	100%	143	100%	4330	100%	31	100%	31	100%	62	100%
Month	January	20	0.5%	1	0.7%	21	0.5%	1	3.2%	0	0.0%	1	1.6%
	February	60	1.4%	4	2.8%	64	1.5%	2	6.5%	0	0.0%	2	3.2%
	March	202	4.8%	1	0.7%	203	4.7%	1	3.2%	1	3.2%	2	3.2%
	April	605	14.4%	12	8.4%	617	14.2%	2	6.5%	1	3.2%	3	4.8%
	May	1061	25.3%	28	19.6%	1089	25.2%	5	16.1%	1	3.2%	6	9.7%
	June	1029	24.6%	30	21.0%	1059	24.5%	3	9.7%	2	6.5%	5	8.1%
	July	487	11.6%	21	14.7%	508	11.7%	6	19.4%	6	19.4%	12	19.4%
	August	259	6.2%	15	10.5%	274	6.3%	5	16.1%	4	12.9%	9	14.5%
	September	140	3.3%	10	7.0%	150	3.5%	0	0.0%	6	19.4%	6	9.7%
	October	231	5.5%	12	8.4%	243	5.6%	1	3.2%	3	9.7%	4	6.5%
	November	72	1.7%	7	4.9%	79	1.8%	3	9.7%	7	22.6%	10	16.1%
	December	21	0.5%	2	1.4%	23	0.5%	2	6.5%	0	0.0%	2	3.2%
Year	2017	668	16.0%	22	15.4%	690	15.9%	5	16.1%	2	6.5%	7	11.3%
	2018	953	22.8%	20	14.0%	973	22.5%	5	16.1%	0	0.0%	5	8.1%
	2019	740	17.7%	26	18.2%	766	17.7%	8	25.8%	7	22.6%	15	24.2%
	2020	615	14.7%	20	14.0%	635	14.7%	7	22.6%	3	9.7%	10	16.1%
	2021	515	12.3%	26	18.2%	541	12.5%	4	12.9%	8	25.8%	12	19.4%
	2022	696	16.6%	29	20.3%	725	16.7%	2	6.5%	11	35.5%	13	21.0%
Sex	Male	2490	59.5%	89	62.2%	2579	59.6%	17	54.8%	21	67.7%	38	61.3%
	Female	1696	40.5%	54	37.8%	1750	40.4%	14	45.2%	10	32.3%	24	38.7%
Age	<5	495	11.8%	10	7.0%	505	11.7%	0	0.0%	1	3.2%	1	1.6%
	5–10	517	12.4%	14	9.8%	531	12.3%	3	9.7%	0	0.0%	3	4.8%
	10–15	243	5.8%	6	4.2%	249	5.8%	2	6.5%	1	3.2%	3	4.8%
	15–20	176	4.2%	8	5.6%	184	4.3%	2	6.5%	0	0.0%	2	3.2%
	20–25	163	3.9%	6	4.2%	169	3.9%	0	0.0%	0	0.0%	0	0.0%
	25–30	199	4.8%	6	4.2%	205	4.7%	0	0.0%	0	0.0%	0	0.0%
	30–35	184	4.4%	8	5.6%	192	4.4%	1	3.2%	0	0.0%	1	1.6%
	35–40	188	4.5%	6	4.2%	194	4.5%	1	3.2%	1	3.2%	2	3.2%
	40–45	172	4.1%	10	7.0%	182	4.2%	1	3.2%	1	3.2%	2	3.2%
	45–50	221	5.3%	9	6.3%	230	5.3%	1	3.2%	2	6.5%	3	4.8%
	50–55	265	6.3%	7	4.9%	272	6.3%	6	19.4%	2	6.5%	8	12.9%
	55–60	293	7.0%	15	10.5%	308	7.1%	3	9.7%	8	25.8%	11	17.7%
	60–65	289	6.9%	12	8.4%	301	7.0%	7	22.6%	5	16.1%	12	19.4%
	65–70	272	6.5%	13	9.1%	285	6.6%	2	6.5%	3	9.7%	5	8.1%
	70–75	243	5.8%	6	4.2%	249	5.8%	0	0.0%	3	9.7%	3	4.8%
	75+	266	6%	7	4.9%	273	6.3%	2	6.5%	4	12.9%	6	9.7%

LB: Lyme borreliosis; TBE: tick-borne encephalitis; * *ED-bite* events; ** *ED-TBD* events.

**Table 2 ijerph-20-06822-t002:** Hospitalisation rates for LB and TBE in the district and subareas of interest. Local Health Authority 8 (Azienda ULSS Berica), Vicenza, Veneto Region, Italy, 2017–2022.

Area	LB-Related Hospitalisation Rate (per 100,000/Year)	TBE-Related Hospitalisation Rate (per 100,000/Year)
North-west subarea	2.08	1.84
South-west subarea	1.60	0.00
Rest of LHA district	0.34	0.46
Entire LHA district	0.99	0.78

LB: Lyme borreliosis; TBE: tick-borne encephalitis; LHA: local health authority.

## Data Availability

The data presented in this study are available on request from the corresponding author.

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
