# Peer review of "Emergency Department Syndromic Surveillance to Monitor Tick-Borne Diseases: A 6-Year Small-Area Analysis in Northeastern Italy"

_ijerph, 2023, doi:10.3390/ijerph20196822_

Round 1

Reviewer 1 Report

The paper reports the results of an evaluation of a supplementary surveillance system for Tick-Borne diseases in North-Eastern Italy. The topic is of high interest for public health and the methodology is scientifically sound. Some remarks to be taken into consideration to improve interest and readability:

- In the Introduction the current statutory system for TBD surveillance in Italy should be described: is there any? is it functioning? In the discussion the authors mention an ISS bulletin, but without describing the the system behind it. 

- The EDSyS system should support preventive interventions. It should be suggested by the authors how a similar use could look like. For instance, is there a threshold of tick bytes related ED visit that can be used to alert the population that the "tick season" has started? Warning population about it could be beneficial. The granularity of information that such system is able to provide could help health authorities to set up specific alerts (like orange or red level warning) in specific geographic areas at the exact time when the tick activity is increasing.

- The authors barely mention vaccine prevention of TBE. I believe that such system could be very useful in identifying the specific areas where TBE vaccination should be recommended according to TBE risk.

Author Response

The paper reports the results of an evaluation of a supplementary surveillance system for Tick-Borne diseases in North-Eastern Italy. The topic is of high interest for public health and the methodology is scientifically sound. Some remarks to be taken into consideration to improve interest and readability:

- In the Introduction the current statutory system for TBD surveillance in Italy should be described: is there any? is it functioning? In the discussion the authors mention an ISS bulletin, but without describing the the system behind it.

---Thank you for your comments. A brief description of the surveillance in Italy has been added in the introduction section. All changes in the manuscript are highlighted in green.

- The EDSyS system should support preventive interventions. It should be suggested by the authors how a similar use could look like. For instance, is there a threshold of tick bytes related ED visit that can be used to alert the population that the "tick season" has started? Warning population about it could be beneficial. The granularity of information that such system is able to provide could help health authorities to set up specific alerts (like orange or red level warning) in specific geographic areas at the exact time when the tick activity is increasing.

----Given the innovative approach, such a sharp threshold has not yet been established. However, we agree with you about the potential benefits of disseminating messages informed on this basis. With this in mind, we have added a short paragraph reporting your observation in the discussion.

- The authors barely mention vaccine prevention of TBE. I believe that such system could be very useful in identifying the specific areas where TBE vaccination should be recommended according to TBE risk.

---In line with your comment, a few passages on the importance and possibility of recourse to TBE vaccination have been added to the discussion.

Reviewer 2 Report

Dear Authors,

Presented manuscript concentrated on emergency department syndromic surveillance system as a source of complementary data to monitor the risk of tick-borne diseases in Italy. This manuscript is a well-written and presents useful data for the reader.

Minor comments:

Line 119: Please add the abbreviation of LHA: Local Health Authority (LHA)

Figure 1 (Line 172-173): Please check and confirm the data series in a chart. In case of the gray line should be ‘Bite-related ED visits (ED-bite)’ as described on Figure 2, not  ‘Total ED visits’ – please check. Description of Figure 1 In line 174 indicates ‘Number of bite-related ED visits’. However, if you present the total ED visits, I suggest to change for bite-related ED visits.

Line 235-237: Please add reference [37] at the end of sentence ‘Newitt and colleagues described the highest incidence of bite-related consultations in the 45-64 age group and in the 0-15 age group, with some differences in the use of primary care and hospital care.’

Line 350-354: Please, rearrange the sentence because it is exactly the same as in work of cited Marx and colleagues  ‘…can provide timely information that might predict temporal and geographic risk for exposure to tickborne diseases and guide actionable public health messaging such as avoiding tick habitats, wearing repellent consistently when outdoors, and performing regular tick checks during times of increased tick bite risk.’

Line 244-246: Please, rearrange the sentence as above. From Daly et al. Tick bite and Lyme disease-related emergency department encounters in New Hampshire, 2010-2014. Zoonoses Public Health. 2017, 64(8):655-661. doi: 10.1111/zph.12361: ‘This observation is consistent with other studies showing emergence of overwintering adult ticks and nymphs in the spring and presence of adult ticks in the fall (Falco et al., 1999; Schulze & Jordan, 1996; Simmons, Shea, Myers-Claypole, Kruise, & Hutchinson, 2015; Xu et al., 2016). Also, I suggest citing only the work of Daly and colleagues, as your references from 33 to 36 were cited only once in the manuscript, after the above sentence.  I suggest arranging the sentences in lines 247-249 (in Daly et al.: … ED encounters showed a single peak in the summer thought to result from increased exposure to the nymphal stage of the tick, which emerges in the spring) and in lines 254-255 (in Daly et al. … These trends in tick bite-related ED data could provide an early indication of LD risk and influence timing of tick-borne illness prevention messaging.) and add the reference [18].

Author Response

Dear Authors,

Presented manuscript concentrated on emergency department syndromic surveillance system as a source of complementary data to monitor the risk of tick-borne diseases in Italy. This manuscript is a well-written and presents useful data for the reader.

---Dear Reviewer, thank you for all your comments below and appreciation also. All changes in the manuscript are highlighted in green.

Minor comments:

Line 119: Please add the abbreviation of LHA: Local Health Authority (LHA)

---The abbreviation “LHA” has been added.

Figure 1 (Line 172-173): Please check and confirm the data series in a chart. In case of the gray line should be ‘Bite-related ED visits (ED-bite)’ as described on Figure 2, not  ‘Total ED visits’ – please check. Description of Figure 1 In line 174 indicates ‘Number of bite-related ED visits’. However, if you present the total ED visits, I suggest to change for bite-related ED visits.

---Figure 1 has been modified as suggested above. Figure labels, captions and text in the manuscript are now all consistent.

Line 235-237: Please add reference [37] at the end of sentence ‘Newitt and colleagues described the highest incidence of bite-related consultations in the 45-64 age group and in the 0-15 age group, with some differences in the use of primary care and hospital care.’

---Reference added.

Line 350-354: Please, rearrange the sentence because it is exactly the same as in work of cited Marx and colleagues  ‘…can provide timely information that might predict temporal and geographic risk for exposure to tickborne diseases and guide actionable public health messaging such as avoiding tick habitats, wearing repellent consistently when outdoors, and performing regular tick checks during times of increased tick bite risk.’

---The sentence has been rearranged as suggested.

Line 244-246: Please, rearrange the sentence as above. From Daly et al. Tick bite and Lyme disease-related emergency department encounters in New Hampshire, 2010-2014. Zoonoses Public Health. 2017, 64(8):655-661. doi: 10.1111/zph.12361: ‘This observation is consistent with other studies showing emergence of overwintering adult ticks and nymphs in the spring and presence of adult ticks in the fall (Falco et al., 1999; Schulze & Jordan, 1996; Simmons, Shea, Myers-Claypole, Kruise, & Hutchinson, 2015; Xu et al., 2016). Also, I suggest citing only the work of Daly and colleagues, as your references from 33 to 36 were cited only once in the manuscript, after the above sentence.

----The sentence has been rearranged and redundant references removed.

I suggest arranging the sentences in lines 247-249 (in Daly et al.: … ED encounters showed a single peak in the summer thought to result from increased exposure to the nymphal stage of the tick, which emerges in the spring) and in lines 254-255 (in Daly et al. … These trends in tick bite-related ED data could provide an early indication of LD risk and influence timing of tick-borne illness prevention messaging.) and add the reference [18].

---The sentences have been rearranged and partly deleted because the same concept is explained a few lines later.